# Medication-Related Problems in Older People with Multimorbidity in Catalonia: A Real-World Data Study with 5 Years’ Follow-Up

**DOI:** 10.3390/jcm10040709

**Published:** 2021-02-11

**Authors:** Amelia Troncoso-Mariño, Albert Roso-Llorach, Tomás López-Jiménez, Noemí Villen, Ester Amado-Guirado, Sergio Fernández-Bertolin, Lucía A. Carrasco-Ribelles, Josep Ma Borras, Concepción Violán

**Affiliations:** 1Medicines Area and Pharmacy Service, Barcelona Territorial Management, Institut Català de la Salut, 08015 Barcelona, Spain; atroncoso@gencat.cat (A.T.-M.); nvillenr.bcn.ics@gencat.cat (N.V.); eamado@gencat.cat (E.A.-G.); 2Department of Clinical Sciences, University of Barcelona and IDIBELL, L’Hospitalet de Llobregat, 08908 Barcelona, Spain; jmborras@iconcologia.net; 3Fundació Institut Universitari per a la Recerca a l’Atenció Primària de Salut Jordi Gol i Gurina (IDIAPJGol), 08007 Barcelona, Spain; aroso@idiapjgol.org (A.R.-L.); tlopez@idiapjgol.org (T.L.-J.); sfernandez@idiapjgol.org (S.F.-B.); luciacarrascoribelles@gmail.com (L.A.C.-R.); 4Departament de Pediatria, Obstetricia i Ginecologia i Medicina Preventiva, Universitat Autònoma de Barcelona, 08193 Bellaterra (Cerdanyola del Vallès), Spain; 5Departament de Teoria del Senyal i Comunicacions, Universitat Politècnica de Catalunya, 08034 Barcelona, Spain; 6Research Support Unit Metropolitana Nord, Fundació Institut Universitari per a la Recerca a l’Atenció Primària de Salut Jordi Gol i Gurina (IDIAPJGol), Mataró, 08303 Barcelona, Spain; 7Concepción Violán, Research Support Unit Metropolitana Nord, Fundació Institut Universitari per a la Recerca a l’Atenció Primària de Salut Jordi Gol i Gurina (IDIAPJGol). Mare de Déu de Guadalupe 2, planta 1ª, Mataro, 08303 Barcelona, Spain

**Keywords:** multimorbidity, polypharmacy, drug interactions, duplicate therapy, contraindicated drugs, inappropriate prescribing, primary health care, elderly

## Abstract

Aging, multimorbidity, and polypharmacy are associated with medication-related problems (MRPs). This study aimed to assess the association that multimorbidity and mortality have with MRPs in older people over time. We followed multimorbid, older (65–99 years) people in Catalonia from 2012 to 2016, using longitudinal data and Cox models to estimate adjusted hazard ratios (HR). We reviewed electronic health records to collect explanatory variables and MRPs (duplicate therapy, drug–drug interactions, potentially inappropriate medications (PIM), and contraindicated drugs in chronic kidney disease (CKD) or liver disease). There were 723,016 people (median age: 74 years; 58.9% women) who completed follow-up. We observed a significant (*p* < 0.001) increase in the proportion with at least one MRP (2012: 66.9% to 2016: 75.5%); contraindicated drugs in CKD (11.1 to 18.5%) and liver disease (3.9 to 5.3%); and PIMs (62.5 to 71.1%), especially drugs increasing fall risk (67.5%). People with ≥10 diseases had more MRPs (in 2016: PIMs, 89.6%; contraindicated drugs in CKD, 34.4%; and in liver disease, 9.3%). All MRPs were independently associated with mortality, from duplicate therapy (HR 1.06; 95% confidence interval (CI) 1.04–1.08) to interactions (HR 1.60; 95% CI 1.54–1.66). Ensuring safe pharmacological treatment in elderly, multimorbid patient remains a challenge for healthcare systems.

## 1. Introduction

More than a third of the population over the age of 50 years in Europe, and 39% in Spain, presents multimorbidity, defined as having at least two chronic diseases [1]. Age is an important determinant of multimorbidity, with people’s risk increasing as they grow older [2,3]. Another factor frequently associated with multimorbidity is polymedication (routine use of five or more medications), which often arises from the application of disease-centered clinical practice guidelines, resulting in patients receiving all the indicated treatments for a single pathology, without regard to the other diseases that they may have [4]. Furthermore, treatment indications are seldom adjusted by age group. This aspect, together with the fragmentation of patient care at different healthcare levels, makes it difficult to obtain a holistic understanding of which treatments should be recommended [5].

Medication-related problems (MRPs) are defined as any event or situation wherein medicines may actually or potentially impede the achievement of positive health outcomes [6]. Numerous studies have aimed to quantify MRPs in the hospital setting; however, these problems are also prevalent in primary health care (PHC), where drug treatments are among the most common resources deployed. Polymedication increases the risk of MRPs [7]. As the promotion of patient safety has emerged as a priority for health systems worldwide, in 2017 the World Health Organization (WHO) set a target to cut the global level of severe, avoidable harms related to medications in half by 2022 [8].

The APEAS study, a cross-sectional observational study in 48 PHC centers in 16 semi-autonomous regions of Spain, aimed to assess the frequency of adverse events and the factors contributing to their appearance, their severity, and their amenability to prevention [9]. The authors found that medications are a causal factor in 48% of patient safety incidents in PHC, and of these, 58% are preventable. Advanced age, polymedication, and multimorbidity are among the risk factors commonly related to MRPs [10], the most serious of which may precipitate an emergency room visit, hospital admission, or death [11,12].

Older patients have certain characteristics that make them especially vulnerable to MRPs. Multimorbidity, polymedication, and changes in drug pharmacodynamics and pharmacokinetics affect most people over the age of 65 years [13,14]. Despite the increasing prevalence of multimorbidity and its cascading impacts on MRPs, patients, and health systems, we are not aware of any longitudinal studies that analyze MRP-related risks in older people with multimorbidity. The aims of this study were to evaluate the relationship between MRPs and multimorbidity in patients over 65 years of age and receiving treatment in primary health care (PHC) in Catalonia (Spain) from 2012 to 2016 and to study five-year mortality related to MRPs.

## 2. Materials and Methods

### 2.1. Design, Setting, and Study Population

This retrospective study with longitudinal data took place in Catalonia, a Mediterranean region of Spain with 7,675,217 inhabitants in 2019 [15] and a decentralized, universal health coverage model financed by tax revenue and encompassed under the Spanish National Health System. The Catalan Health Institute oversees 285 PHC centers, serving 5,501,784 patients (74% of the population); other providers manage the remaining PHC centers [16].

We included people who: were aged 65 to 99 years on 31 December 2011, lived at least until 31 December 2012 (index date), presented multimorbidity in 2012 or 2016, and made at least one visit to PHC during the five-year study period (2012–2016). The cohort was closed to new entries; attrition was the result of mortality or loss to follow-up due to transfer to different health systems.

### 2.2. Dataset

The Catalan Health Institute’s Information System for Research in Primary Care (SIDIAP) stores information from electronic health records (EHR), recorded in PHC centers since 2006 [17]. The SIDIAP database contains anonymized EHRs from PHC and secondary care, with longitudinal information on demographics, socioeconomic status, diagnoses, symptoms, and prescriptions.

The medication database covers all drugs that are dispensed, subsidized, and billed by the national health service. To identify problems associated with the most relevant medications, we included only drugs for systemic use, excluding medications with local effects (e.g., topical drugs). This database does not cover medication administered in hospitals or dispensed by hospital pharmacies, nor drugs that are not subsidized through public healthcare services.

### 2.3. Variables

All variables were obtained directly from the SIDIAP database [18].

#### 2.3.1. Chronic Diseases and Multimorbidity

The SIDIAP database codes diseases using the International Classification of Diseases, 10th revision (ICD-10). We considered multimorbidity as the presence of two or more chronic diseases, defined using 60 selected groups of chronic diseases described in the Swedish National Study of Aging and Care in Kungsholmen (SNAC-K) [19], which was based on ICD-10 diagnostic codes, along with certain clinical, laboratory, and medication-related parameters. Patients were classified into four multimorbidity categories according to the number of pathologies they had: 0 to 1; 2 to 4; 5 to 9; and 10 or more chronic diseases. The number of different chronic diseases per patient was collected at baseline (2012) and study end (2016).

#### 2.3.2. Drugs

We obtained data for drug exposure from the Pharmacy Invoice Registry. This registry records drugs prescribed by PHC and hospital physicians. The drugs were categorized according to the fourth and fifth levels of the Anatomical Therapeutic Chemical Classification System (ATC) [20], facilitating analysis and interpretation. When the person used three packages of the drug per year during the study period, this was considered chronic use. Packages were defined as the pre-prepared packet of drugs constituting the sales unit of medications available in the pharmacy; these frequently contain the number of doses needed for one month of the indicated treatment. Drugs were coded as dichotomous variables [21], and we classified patients into five categories according to the number of different prescription drugs they were on: 0; 1; 2 to 4; 5 to 9; 10 or more.

#### 2.3.3. Kidney Function

Impaired kidney function was defined by one of the two following parameters:(a)Glomerular filtration rate (GFR): using the MDRD-4 IDMS equation, kidney function was determined by estimating the GFR [22]. GFR values of less than 60 mL/min/1.73 m^2^ indicated impaired kidney function.(b)ICD-10 codes for chronic kidney disease (CKD) were taken from SNAC-K criteria [23].

#### 2.3.4. Liver Function

Impaired liver function was defined by one of the two following parameters:(a)Abnormal liver function values: alkaline phosphatase (ALP) > 2 × 129 IU/L; alanine transaminase (ALT) > 5 × 41 IU/L (men) or ALT > 5 × 33 IU/L (women); and gamma-glutamyl transpeptidase (GGT) > 61 IU/L [24].(b)ICD-10 codes for chronic liver disease were taken from SNAC-K criteria [23].

#### 2.3.5. Other Variables

Additional variables collected at baseline and/or study end were sociodemographic variables (age at baseline, gender, and socioeconomic status according to MEDEA index quintiles from least to most deprived) [25] along with the number of total visits to PHC.

#### 2.3.6. Medication-Related Problems

We analyzed the following MRPs: duplicate therapy, drug–drug interactions, potentially inappropriate medications (PIMs) in people aged 65 years or older, contraindicated drugs in CKD, and contraindicated drugs in liver disease. Analysis of the MRPs began by first building tables with medicines and combination drug treatments associated with potential safety concerns:-Duplicate therapy: prescription of two or more drugs that have the same pharmacological activity. We included duplicate therapies that posed an important clinical risk, according to professional consensus. Our study did not consider combinations of active principles with the same pharmacological action that physicians used to achieve a synergistic effect or to adjust doses [23].-Drug–drug interaction: when one drug’s activity or effect alters the action of another. We focused on interactions that were life-threatening due to therapeutic failure or toxicity, identifying interactions with the highest level of severity (contraindication) from the *Thesaurus des interactions médicamenteuses* from France’s *Agence Nationale de Sécurité du Médicament et des Produits de Santé* (ANSM) [26] and comparing that information with a second source [27,28] or contrasting it with professional consensus [23].-Contraindicated drugs in CKD: we identified these drugs from the Catalan Health Department’s consensus recommendation for patients with CKD [23,29].-Contraindicated drugs in liver disease: the Spanish College of Pharmacists database was used to identify these [23,30].-PIMs in older people (≥65 years old): when the risk of adverse events associated with the drug exceeded the expected clinical benefits, and there was no clear scientific evidence pointing to a specific indication or supporting its cost-effectiveness. We primarily used the STOPP/START criteria [31], which was complemented with Beers’ criteria [32], PRISCUS, and updates from other sources [33,34,35,36,37,38,39,40]. Different definitions of PIMs exist: Durán et al. defined them as drug combinations with a clinically relevant anticholinergic effect [38], while other authors have included drugs that increase the risk of falling or affect the QT interval with known risk, antiulcer agents administered without considering gastroprotection, and other drugs that are inadvisable for older people or patients needing gastroprotection [23].

### 2.4. Ethics

The Clinical Research Ethics Committee of the Fundació Institut Universitari per a la Recerca a l’Atenció Primària de Salut Jordi Gol i Gurina (IDIAPJGol) (Protocol No: P17/080) approved the study protocol. Data were fully anonymized, and the confidentiality of the EHR was guaranteed at all times in accordance with national and international law.

### 2.5. Statistical Analysis

We used descriptive statistics to summarize the dataset, expressing categorical variables as absolute and relative frequencies and continuous variables as mean (standard deviation, SD) or median (interquartile range, IQR), as appropriate. Differences between multimorbidity groups that completed the follow-up at each study time point were assessed using the chi-square test or the Mann–Whitney U test. The McNemar and Wilcoxon tests were applied to compare prevalence of MRPs and use of PIMs between multimorbidity groups at baseline and at five years’ follow-up. The effect of age on the prevalence of MRPs and the use of PIMs were also studied, using data from 2012. For this purpose, two logistic regressions were performed for each MRP and PIM, one only with age and the other by multimorbidity group at baseline, adjusting for age. Multinomial logistic regressions were used to study the influence on the use of PIMs in terms of their anticholinergic effect–anticholinergic load, in both cases.

To estimate mortality hazard for each MRP (duplicate therapy, drug–drug interactions, contraindicated drugs in CKD, contraindicated drugs in liver disease, and PIMs), we fitted Cox proportional-hazards regression models. Length of follow-up was the time elapsed from the index date to all-cause death. For the survival analysis, we followed patients until loss to follow-up (censored) or end of observation. Hazard ratios (HRs) and 95% confidence intervals (CIs) were adjusted for age, sex, socioeconomic status (MEDEA index), and level of multimorbidity at baseline. The proportional hazard assumption was assessed by means of the Schoenfeld residuals. We employed multiple imputation to minimize any selection bias stemming from missing values for MEDEA (7%) and chained equations to obtain seven imputed datasets. The multiple imputation datasets were incorporated into the final models according to Rubin’s rules for combining effect estimates and standard errors, which allowed for some uncertainty related to missing data.

Analyses were performed using SPSS for Windows (version 25, SPSS Inc., New York, NY, USA), Stata 15 Stata/MP (version 15 for Windows, Stata Corp. LP, College Station, TX, USA), and R (version 4.0.3, R Foundation for Statistical Computing, Vienna, Austria). *p* values of less than 0.05 were considered significant.

## 3. Results

A total of 916,619 people aged 65 years or older were included in the database; 853,085 met the multimorbidity criteria, and 723,016 of these completed five years of follow-up, as shown in Figure 1. This population’s median age was 74 years (IQR 68 to 79), and 58.9% were women. At five years, there was an increase in the proportion of people taking 5 to 9 drugs (39.8% vs. 43.1%, *p* < 0.001) and more than 10 drugs (12.6% vs. 15.4%, *p* < 0.001). Likewise, a higher percentage of patients presented altered liver or kidney function and visited PHC 10 times or more per year, as shown in Table 1.

MRPs were highly prevalent in the study population and increased over time: 66.9% of the patients had at least one MRP in 2012, compared to 75.5% in 2016, as shown in Figure 2a. Changes in the type of MRP varied over the study period, with a drop in duplicate therapies (11.3% vs. 6.2%) and drug–drug interactions (0.9% vs. 0.5%). In contrast, the use of contraindicated medicines in CKD and liver disease as well as PIMs had risen by study end, as shown in Figure 2b.

In all cases, patients’ MRPs increased with the number of chronic diseases; those with 10 or more had the highest prevalence of MRPs both at baseline and study end. At the end of follow-up, 89.6% of this subgroup were taking PIMs; 34.4%, contraindicated drugs with CKD; and 10.1%, duplicate therapy; smaller proportions had other MRPs, as shown in Table 2. In addition, after adjusting for age, the association between MRPs and multimorbidity was still significant, as shown in Appendix A. The most common PIMs in the whole study population at five-year follow-up were drugs increasing the risk of falling (67.5%); this figure was higher in the subgroup with 10 or more comorbidities, as shown in Table 3. In 2016, 48.4% of the study population was taking at least two PIMs, along with 8.3% who were taking at least two contraindicated drugs in CKD, and 2.9%, at least two contraindicated drugs in liver disease, as shown in Appendix A.

The most common contraindicated drugs in CKD were metformin (4.3%), hydrochlorothiazide (both as monotherapy (2.8%) and in fixed-dose combination with enalapril (2.1%)), and citalopram (1.6%), as shown in Appendix A. In liver disease, the main contraindicated drugs were simvastatin (1.9%), furosemide (1.2%), and metformin (1.0%), as shown in Appendix A. The most frequent PIMs were omeprazole, used without gastroprotective criteria (23.3%); lorazepam, due to the increased risk of falling (14.8%); and bisoprolol, in patients concomitantly taking anticholinesterase, due to the risk of cardiac conduction failure, syncope, or heart damage (11.6%), as shown in Appendix A.

People with MRPs were at higher risk of death—especially in the case of drug–drug interactions (HR 1.62, 95% CI 1.55 to 1.70), followed by patients taking contraindicated drugs in liver disease (HR 1.59, 95% CI 1.56 to 1.63) and those on PIMs (HR 1.31, 95% CI 1.29 to 1.32), as shown in Table 4.

## 4. Discussion

Our results show that MRPs, and especially PIMs, were very prevalent in multimorbid older people both at baseline and at study end, especially in patients with the most comorbidities. Presenting one or more MRPs was associated with a higher risk of mortality at five years’ follow-up.

Multimorbidity is associated with the prescription of multiple medications, boosting the risk of exposure to unnecessary drugs and the likelihood of duplicate therapies or interactions, among other MRPs [41]. Aging is associated with diminished renal and liver clearance, primarily due to the reduced blood flow and hepatocyte mass as well as sclerotic changes in the glomeruli [14]. These factors set off changes in drug metabolism and excretion, which can make it necessary to adjust drug dosing or use alternative treatments that are not contraindicated in these situations.

A variety of terminology is used to conceptualize different events or circumstances related to administering medication that could actually or potentially interfere in achieving desired health outcomes, making it extremely difficult to compare the results obtained in different studies [42].

A singular contribution of this study is that it analyzes the prevalence and five-year evolution of MRPs in a large cohort of multimorbid older people. To our knowledge, no other studies have reported longitudinal data or trends associated with duplicate therapy or contraindicated drugs in CKD or liver disease after a follow-up period. We observed a decrease in the proportion of patients on duplicate therapy, from 11.3% at baseline to 6.2% at five years, which is more similar to the 4.8% reported in cross-sectional studies [43]. Similarly, we saw a decrease in drug–drug interactions; this result stands in contrast with other studies that not only reported an upward trend, but also a higher overall prevalence of this problem [44]. These differences may be because we excluded over-the-counter medications and dietary supplements from the analysis, instead analyzing only formally contraindicated interactions. In addition, in 2012 the Catalan Health Institute implemented a warning system for clinical histories in PHC to prevent medication-related errors, including interactions and duplications. This policy could have contributed to the decrease in these MRPs over the five-year follow-up, although a specific study investigating this relationship would be needed to confirm this hypothesis.

On the other hand, our results show an increase in the use of contraindicated drugs in CKD, from 11.1% at baseline to 18.5% at five years. Cross-sectional studies show wide variations (13 to 80.5%) in the proportion of patients with CKD who are prescribed inappropriate drugs in ambulatory care [45]. As in our study, metformin is among the most common medications used inappropriately in these patients [45]. Thus, automated warnings to support dose adjustment in electronic prescription systems may favor appropriate drug use in patients with CKD.

We did not identify PHC studies that analyzed contraindicated drugs in liver disease in older patients. The influence of these pathologies on pharmacokinetics and pharmacodynamics is complex, and currently there is no quantitative method to calculate dose adjustments in these patients. These challenges mean that recommendations on drug use are imprecise, and few studies have tried to quantify this type of MRP. However, we observed a high prevalence of this problem, so it needs to be considered when prescribing drugs in older people.

With regard to the number of PIMs in the elderly patient population, the literature reports a wide range and divergent trends. Some studies reported no significant changes over time, while others found, as we did, that this type of MRP tends to increase over follow-up [46,47]. Notably, the proportion of patients taking PIMs was higher in our study (ranging from 62.5 to 71.1%) than in others. Diverse factors could play a role in these differences; for example, variations in the criteria used to identify a PIM or in the characteristics of included patients. Regarding the criteria for defining a PIM, we used a comprehensive review, which could have influenced the number of PIMs identified [23].

In our study, we observed that older, multimorbid patients who presented MRPs carry a higher risk of mortality, and the magnitude of this association is highest in those with drug–drug interactions. Mortality associated with drug-related adverse events has mainly been analyzed in the hospital setting, where it represents 0.15% of all inpatient deaths—double the rate as in the rest of admitted patients [48,49]. Our study corroborates the data reported in a hospital setting from the PHC perspective, highlighting the impact of MRPs for both patients and the health system.

This study is based on a large, high-quality database containing PHC records that are representative of the multimorbid population in Catalonia [18]. In addition, we used validated, clinically driven methodology to measure chronic diseases and polypharmacy, which allows a standardized evaluation of chronic diseases in the European Union [19].

Nevertheless, our study also has some limitations. First of all, we considered only the medications used chronically (≥3 packages dispensed per year), thus excluding medications used to treat acute pathologies from the analysis. Secondly, the SIDIAP database only collects information on medications prescribed by PHC and hospital-based physicians that are dispensed in community pharmacies and covered under the national health system, with no data on drugs dispensed in hospital, over-the-counter medications, and those not financed by the health system. These two limitations could have caused us to underestimate the frequency of MRPs [50]. Finally, due to the study methodology, it was not possible to report patients’ adherence to treatment.

People with multimorbidity usually receive care from several prescribers, including physicians in PHC and in hospitals [51]. Polymedication is often the result of fragmentation of the health system, together with the application of clinical practice guidelines that focus on a single pathology rather than holistically taking into account the complexity of patients with multimorbidity [4]. Primary care physicians and pharmacists have an important role in reconciling different prescriptions during care transitions and in reviewing treatments in patients with an advanced age and multimorbidity. These professionals can help to align drug regimens with individuals’ desired outcomes, weighing the risk–benefit balance of treatments recommended in clinical practice guidelines.

The data on MRPs obtained in this study provide information that could inform the design of programs for preventing MRPs in healthcare institutions. Interventions aimed at avoiding drug-related adverse events could include integrating decision-making aids into EHRs or implementing professional training.

## 5. Conclusions

The high prevalence of MRPs and their possible impact on mortality in elderly patients with multimorbidity suggest that MRPs are a widespread public health problem. Our results show the impact of MRPs when patients have more comorbidities, affecting both patients’ health and healthcare services.

## Figures and Tables

**Figure 1 jcm-10-00709-f001:**
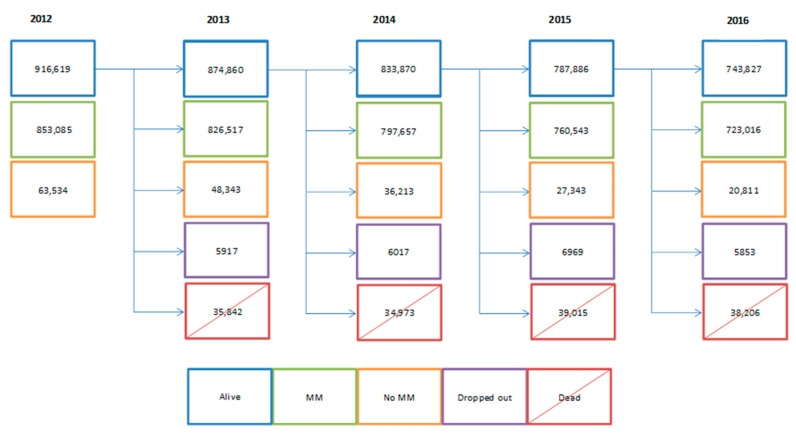
Longitudinal flow chart of patients meeting initial inclusion criteria, without consideration for the multimorbidity criterion (2012–2016; N = 916,619 people).

**Figure 2 jcm-10-00709-f002:**
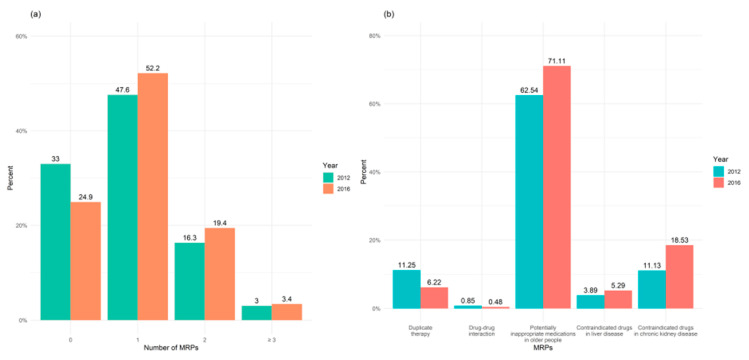
(**a**) Frequencies and (**b**) types of medication-related problems (MRPs) in older people with multimorbidity at baseline (2012) and at end of study (2016) (N = 723,016).

**Table 1 jcm-10-00709-t001:** Descriptive characteristics of older people by multimorbidity groups at baseline and at the end of the study (N = 723,016).

Variables	N Comorbidities							
	2–4 Diseases N = 137,799		5–9 Diseases N = 393,672		≥10 Diseases N = 191,545		Total N = 723,016	
	*n* (%) *		*n* (%) *		*n* (%) *		*n* (%) *	
Sex, women, *n* (%)	73,137 (53.1)		229,870 (58.4)		123,014 (64.2)		426,021 (58.9)	
Age, median (IQR)	71.00 [67.00,76.00]		73.00 [68.00;79.00]		76.00 [71.00;80.00]		74.00 [68.00;79.00]	
Rural	30,094 (22.5)		78,760 (20.6)		31,752 (17.2)		140,606 (20.1)	
MEDEA Index †								
Q1	26,608 (19.9)		63,322 (16.6)		27,986 (15.1)		117,916 (16.8)	
Q2	22,006 (16.5)		62,570 (16.4)		30,182 (16.3)		114,758 (16.4)	
Q3	20,866 (15.6)		63,789 (16.7)		31,480 (17.0)		116,135 (16.6)	
Q4	19,243 (14.4)		61,187 (16.0)		32,159 (17.4)		112,589 (16.1)	
Q5	14,944 (11.2)		52,323 (13.7)		31,567 (17.1)		98,834 (14.1)	
Number of drugs	**2012**	**2016**	**2012**	**2016**	**2012**	**2016**	**2012**	**2016**
0	34,937 (25.4)	26,903 (19.5)	32,578 (8.3)	16,833 (4.3)	7104 (3.7)	1918 (1.0)	74,619 (10.3)	45,654 (6.3)
1	21,226 (15.4)	21,234 (15.4)	24,267 (6.2)	19,235 (4.9)	2774 (1.5)	1661 (0.9)	48,267 (6.7)	42,130 (5.8)
2–4	57,435 (41.7)	62,554 (45.4)	136,887 (34.8)	130,158 (33.1)	26,751 (14.0)	19,860 (10.4)	221,073 (30.6)	212,572 (29.4)
5–9	22,892 (16.6)	25,954 (18.8)	169,574 (43.1)	191,213 (48.6)	95,389 (49.8)	94,522 (49.4)	287,855 (39.8)	311,689 (43.1)
≥10	1309 (1.0)	1154 (0.8)	30,366 (7.7)	36,233 (9.2)	59,527 (31.1)	73,584 (38.4)	91,202 (12.6)	110,971 (15.4)
Number of visits								
0	13,639 (9.9)	10,299 (7.5)	12,374 (3.1)	5901 (1.5)	3147 (1.6)	858 (0.5)	29,160 (4.0)	17,058 (2.4)
1	11,341 (8.2)	10,419 (7.6)	10,723 (2.7)	9130 (2.3)	1523 (0.8)	1276 (0.7)	23,587 (3.3)	20,825 (2.9)
2–4	38,129 (27.7)	38,161 (27.7)	57,257 (14.5)	52,043 (13.2)	10,123 (5.3)	8725 (4.6)	105,509 (14.6)	98,929 (13.7)
5–9	45,344 (32.9)	46,684 (33.9)	128,738 (32.7)	123,262 (31.3)	37,760 (19.7)	32,163 (16.8)	211,842 (29.3)	202,109 (28.0)
≥10	29,346 (21.3)	32,236 (23.4)	184,580 (46.9)	203,336 (51.7)	138,992 (72.6)	148,523 (77.5)	352,918 (48.8)	384,095 (53.1)
CKD	6809 (4.9)	10,907 (7.9)	53,110 (13.5)	96,875 (24.6)	49,182 (25.7)	88,954 (46.4)	109,101 (15.1)	196,736 (27.2)
Chronic liver disease	4495 (3.3)	5228 (3.8)	22,105 (5.6)	26,888 (6.8)	16,883 (8.8)	22,511 (11.8)	43,483 (6.0)	54,627 (7.6)

CKD: chronic kidney disease. Note: All variables of the table showed a significant difference (*p* < 0.001) between groups. * Unless otherwise noted. † MEDEA is a deprivation index measured in quintiles (Q), from Q1 (least deprived) to Q5 (most deprived). Missing values n = 22,178. IQR: interquartile range.

**Table 2 jcm-10-00709-t002:** Medication-related problems in older people by multimorbidity group at baseline and at the end of the study (N = 723,016).

	N Comorbidities								
Medication-Related Problems	2–4 Diseases N = 137,799		5–9 Diseases N = 393,672		≥10 Diseases N = 191,545		Total N = 723,016		
	*n* (%)		*n* (%)		*n* (%)		*n* (%)		
	2012	2016	2012	2016	2012	2016	2012	2016
Duplicate therapy	6853 (5.0)	3565 (2.6)	40,507 (10.3)	22,070 (5.6)	33,997 (17.8)	19,370 (10.1)	81,357 (11.3)	45,005 (6.2)
Drug–drug interactions	396 (0.3)	172 (0.1)	2616 (0.7)	1473 (0.4)	3100 (1.6)	1817 (1.0)	6112 (0.9)	3462 (0.5)
Contraindicated drugs in CKD	3606 (2.6)	5199 (3.8)	37,386 (9.5)	62,808 (16.0)	39,474 (20.6)	65,962 (34.4)	80,466 (11.1)	133,969 (18.5)
Contraindicated drugs in liver disease	2012 (1.5)	2495 (1.8)	13,812 (3.5)	17,959 (4.6)	12,302 (6.4)	17,790 (9.3)	28,126 (3.9)	38,244 (5.3)
Potentially inappropriate medication	49,999 (36.3)	60,353 (43.8)	245,447 (62.4)	282,150 (71.7)	156,710 (81.8)	171,641 (89.6)	452,156 (62.5)	514,144 (71.1)

CKD: chronic kidney disease. Note: All variables of the table showed a significant difference (*p* < 0.001) between groups.

**Table 3 jcm-10-00709-t003:** Potentially inappropriate medications in older people by multimorbidity group at baseline and at the end of the study (N = 514,144).

	N Comorbidities							
Reason for Drug Inappropriateness	2–4 Diseases N = 60,353		5–9 Diseases N = 282,150		≥10 Diseases N = 171,641		Total N = 514,144	
	*n* (%)		*n* (%)		*n* (%)		*n* (%)	
	2012	2016	2012	2016	2012	2016	2012	2016
Anticholinergic effect–anticholinergic load*								
Score = 1	7011 (11.6)	10,910 (18.1)	42,699 (15.1)	61,400 (21.8)	34,069 (19.9)	45,155 (26.3)	83,779 (16.3)	117,465 (22.9)
Score = 2	3614 (6.0)	5104 (8.5)	25,833 (9.2)	32,771 (11.6)	22,269 (13.0)	27,232 (15.9)	51,716 (10.1)	65,107 (12.7)
Score = 3–5	1209 (2.0)	1499 (2.5)	9929 (3.5)	13,023 (4.6)	11,885 (6.9)	15,429 (9.0)	23,023 (4.5)	29,951 (5.8)
Score ≥ 6	39 (0.1)	37 (0.1)	307 (0.1)	315 (0.1)	484 (0.3)	436 (0.3)	830 (0.2)	788 (0.2)
Increase in fall risk	26,366 (43.7)	36,430 (60.4)	145,741 (51.7)	184,449 (65.4)	107,661 (62.7)	125,913 (73.4)	279,768 (54.4)	346,792 (67.5)
Effect on QT interval	4834 (8.0)	6814 (11.3)	34,387 (12.2)	43,157 (15.3)	32,334 (18.8)	37,491 (21.8)	71,555 (13.9)	87,462 (17.0)
Antiulcer agents without criteria for gastroprotection	11,821 (19.6)	15,761 (26.1)	81,766 (29.0)	99,292 (35.2)	65,395 (38.1)	74,812 (43.6)	158,982 (30.9)	189,865 (36.9)
Other drugs not recommended for older people	9396 (15.6)	24,069 (39.9)	72,664 (25.8)	141,221 (50.1)	65,478 (38.2)	109,566 (63.8)	147,538 (28.7)	274,856 (53.5)
Patients needing gastroprotection	70 (0.1)	132 (0.2)	524 (0.2)	683 (0.2)	344 (0.2)	325 (0.2)	938 (0.2)	1140 (0.2)

Note: All variables of the table showed a significant difference (*p* < 0.001) between groups. * Score = 0 Not shown.

**Table 4 jcm-10-00709-t004:** Mortality related to medication-related problems in older people with multimorbidity at baseline year (N = 853,085).

Medication-Related Problems	HR (Crude) 95% CI	Complete Case Analysis HR (Adjusted) * 95% CI	Multiple Imputation HR (Adjusted) * 95% CI
Duplicate therapy	1.14 (1.12–1.16)	1.06 (1.04–1.07)	1.06 (1.04–1.08)
Drug–drug interactions	2.02 (1.95–2.10)	1.62 (1.55–1.70)	1.60 (1.54–1.66)
Contraindicated drugs in chronic kidney disease	1.74 (1.72–1.76)	1.06 (1.05–1.08)	1.08 (1.06–1.09)
Contraindicated drugs in liver disease	1.59 (1.56–1.62)	1.59 (1.56–1.63)	1.54 (1.50–1.57)
Potentially inappropriate medication in older people	1.76 (1.74–1.78)	1.31 (1.29–1.32)	1.30 (1.29–1.32)

HR: hazard ratio. * Adjusted for age, sex, socioeconomic status (MEDEA index), and level of multimorbidity.

## Data Availability

The datasets are not available, since researchers signed an agreement with the Information System for the Development of Research in Primary Care (SIDIAP) concerning confidentiality and security of the dataset, which forbids providing data to third parties. The SIDIAP is subject to periodic audits.

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
