# Peer review of "Medication-Related Problems in Older People with Multimorbidity in Catalonia: A Real-World Data Study with 5 Years’ Follow-Up"

_jcm, 2021, doi:10.3390/jcm10040709_

Round 1
Reviewer 1 Report
The study is an original contribution that could be of interest for many public health researchers and stakeholders: polypharmacy and multimorbidity are highly prevalent in older population. Overall, the research paper is well written. Authors followed a cohort of multimorbid patient (≥2 chronic disease) on a 5 years period (between 2012 and 2016) in a population included nearly all patient of Catalonia. Authors used high quality electronic record to measured medication, multimorbidity and mortality.
Some methodological issues remain to answer to the first objective of the paper (To determine the association between MRPs and multimorbidity from 2012 to 2016). A better statistical methodology and a clarification of this objective would be relevant to study the association between MRPS and multimorbidity. It’s not clear if the authors want to study the association between mRPs and Multimorbidity or the trend in use of medication or MRPs between 2012 and 2016. (see main Issue comment) To answer to this objective, they compared crude MRPs prevalence between 2016 and 2012 with a chi-square test that didn’t allow adjustment for age or number of chronic diseases. Those variables (age and number of chronic diseases) vary during the follow-up study as patient who died during the follow-up may be older and cumulated more diseases than survivors.
The second objective of the paper (to study five years mortality related to MRPs) is clearly defined. Study design, the cox regression and interpretation of results answer adequately to the second objective.
Main Issue
All tables, figures, results and discussion about the comparison between 2012 and 2016 are hazardous. Comparison between MRP crude prevalence in 2012 and 2016 with chi-square test is not an appropriate method to study the trend in prevalence of MRP prevalence among patient with multimorbidity. This methodology didn’t control for an increase in age and the in number of chronic diseases in patients during the follow-up period. Difference in MRPs prevalences observed between 2012 in 2016 may be explain by an increase in age of the cohort and by an increase in number of medical conditions during the study period. A regression model analysis controlling for age using data in 2012 may be more accurate to assessed the association between MRP and multimorbidity. If the authors want to realized a trend analysis, they should justify how their study design allowed to do it.
Minor Issues
Introduction
1- A reference should be include to support the author statement in the sentence “Another factor frequently…..”at lines 44-47
Method
2-At line 94, author should clarify “systemic drugs” terminology. What they mean by °systemic drug”, which drugs were excluded?
3-The author should clarify if number of chronic diseases was assessed at baseline only or at each year of follow-up.
4-In line 107, the authors stated that any patient have 0 or 1 chronic diseases. Did the authors exclude patients with 0 or 1 diseases? More information is needed to explain the situation.
5-At line 113, the authors should be more explicated on the package’s definition. Is it a prescription, a precise number of pills?
6-Authors should clarify if only one or both of two parameters have to be meet to define liver or kidney function.
7-There is no mention by the authors if the proportional hazards assumption is violated or not.
Results
8-Number of decimals of percents should be uniformized when reported (one decimal is probably enough). Some percent have two decimal precision (e.g. 58.92%, line 199; tables 1, 2A, 2B) and others one decimal (e.g. 75.5%, line 205).
9-This comment is in link with comment number 4. In table 1, the authors reported that all patients have at least 2 chronic disease. But in figure 1, authors reported that at least 20,811 patients in 2016 and 63,534 in 2012 have no multimorbidity. Authors should clarify the situation.
10-Table 1 is not easy to understand for the reader. Would it be possible to split into 2 distinct tables? Or maybe authors should report total number of patients(N=) in 2012 and in 2016 in each column of the second part of the table.
Discussion
11-The authors should summarize their major finding in the first paragraph of the discussion in link with their objectives.
11.1-At lines 260-261, authors should bring some clarification to their affirmation. They should explain how their results support their affirmation.
11.2-At lines 262-263, this result didn’t answer to any of the objectives of the papers. Why the authors reported this information?
12-A reference should be include to support the author statement in the sentence “A variety of terminology…..”at lines 272-274.
13-Authors should explain why consideration of only ≥3 packages dispensed per year is a limit.
14-A reference should be include to support the author statement in the sentence “Polymedication is often…..”at lines 335-338
15-Authors did a very strong statement at lines 343-344. Maybe they should qualify their statement.
Conclusion
16-In lines 350-351, they authors stated that their study shows the association between MRPs and age. To my knowledge, none of their results describe an association between MRPs and age.
Author Response
"Please see the attachment."

Reviewer 2 Report
Thank you for an interesting paper. Examining 69 associations between medication related problems and multimorbidity from 2012 to 2016, in patients over 65 years of age and receiving treatment in primary health care (PHC) in Catalonia (Spain), and to study five-year mortality related to medication related problems.
The study is a retrospective longitudinal study with five year follow up including systematically gathered data from a public electronic health record. Variables includes relevant baseline data and multimorbidity, drug exposure, medical-related problems etc. Study limitations are well described in the paper.
The paper is well written, the logic is easy to follow and I have only few comments:
Introduction, page 2, line 59: The text refers to the APEAS study. As the reference is not in English it would be helpful if more information on the study is provided in the text.
Results, page 5, line 216 to 218: The numbers are incorrect. The text should be altered to: "In 2016, 48.38% of the study population who was taking at least one two PIM presented two or more MRPs, along with 8.34% who were taking a two contra-indicated drug in CKD and 2.92%, a two contraindicated drug in liver disease (Table S1)."
Results, page 5, line 216 to 218: I think medical related problems refer to number of comorbidities in table S1. "MRPs" should thus be altered to "number of comorbidities".
Round 2
Reviewer 1 Report
(see comments in green)
